# Parent-Reported Social-Communicative Skills of Children with 22q11.2 Copy Number Variants and Siblings

**DOI:** 10.3390/genes13101801

**Published:** 2022-10-06

**Authors:** Jente Verbesselt, Ellen Van Den Heuvel, Jeroen Breckpot, Inge Zink, Ann Swillen

**Affiliations:** 1Centre for Human Genetics, University Hospitals Leuven, Herestraat 49, 3000 Leuven, Belgium; 2Centre for Developmental Disorders, University Hospital Brussels, Laarbeeklaan 101, 1090 Jette, Belgium; 3Department of Human Genetics, Catholic University Leuven, Herestraat 49, 3000 Leuven, Belgium; 4Research Group Experimental Oto-Rhino-Laryngology (ExpORL), Department of Neurosciences, KU Leuven, Herestraat 49, 3000 Leuven, Belgium; 5MUCLA, Department of Oto-Rhino-Laryngology, Head & Neck Surgery, University Hospitals Leuven, Herestraat 49, 3000 Leuven, Belgium

**Keywords:** copy number variants, 22q11.2 deletion syndrome, 22q11.2 duplication, communication, language, social responsiveness, parental questionnaires

## Abstract

22q11.2 deletion (22q11.2DS) and 22q11.2 duplication (22q11.2Dup) confer risk for neurodevelopmental difficulties, but the characterization of speech-language and social skills in 22q11.2Dup is still limited. Therefore, this study aims to delineate social-communicative skills in school-aged children with 22q11.2Dup (*n* = 19) compared to their non-carrier siblings (*n* = 11) and age-matched children with 22q11.2DS (*n* = 19). Parents completed two standardized questionnaires: the Children’s Communication Checklist (CCC-2), screening speech, language, and social skills, and the Social Responsiveness Scales (SRS-2), assessing deficits in social behavior. Parents report that both children with 22q11.2Dup and 22q11.2DS show more social-communicative deficits than the general population; children with 22q11.2Dup seem to take an intermediate position between their siblings and children with 22q11.2DS. Compared to 22q11.2DS, they demonstrate less frequent and less severe problems, and more heterogeneous social-communicative profiles, with fewer restricted interests and repetitive behaviors. In siblings of 22q11Dup, milder social-communicative difficulties and equally heterogeneous profiles are reported, which might indicate that-in addition to the duplication-other factors such as the broader genetic context play a role in social-communicative outcomes.

## 1. Introduction

Recurrent copy number variants (CNVs) are associated with a significant risk for neurodevelopmental disorders (NDDs), including speech, language, and communication impairments, and social and behavioral difficulties [1,2,3,4,5]. Language and speech are essential for human interaction and communication. Due to their link to and comorbidity with cognition, behavior, and socio-emotional development, they constitute fundamental research topics. Furthermore, they interact with academic achievement and quality of life measures and may be useful for identifying autistic traits [6,7,8,9]. Studies exploring communication skills should address different aspects, such as speech (e.g., articulation of words), structural language (e.g., formulation of sentences), pragmatic language (e.g., use of language in social contexts), and related social components (e.g., social motivation) [10,11].

Recurrent CNVs at chromosomal locus 22q11.2 are among the most common rare genetic disorders that confer significant risk for NDDs across the lifespan, in particular 22q11.2 deletion syndrome (22q11.2DS) and 22q11.2 duplication (22q11.2Dup). To date, social-communicative skills have been thoroughly studied in 22q11.2DS, confirming that both structural and pragmatic language skills may be profoundly affected in receptive as well as expressive language domains [12,13]. Regarding the social and behavioral profile in 22q11.2DS, high rates of autistic features have been described, with an estimated prevalence of autism spectrum disorder (ASD) in 20–42% [14,15,16,17].

The question arises whether a duplication in the same chromosomal region will have a similar impact on social-communicative outcomes. Features of 22q11.2Dup are variable, although it is in general associated with a milder phenotype compared to the 22q11.2DS. Physical features include dysmorphism, transient hearing impairment, nutritional problems, cardiovascular defects, growth retardation, and hypotonia, but all at lower rates compared to the 22q11.2DS [18]. The developmental phenotype is generally characterized by speech-language and motor delays, cognitive impairments, and behavioral problems. Although many case reports describe speech-language delays, impairments, or behavioral problems in patients with 22q11.2Dup, only a few studies have investigated these problems [18,19,20,21,22].

Regarding the social and behavioral profile, Wenger et al. [23] used direct instruments and parental questionnaires, such as the Social Responsiveness Scale (SRS) [24], to characterize the neuropsychiatric functioning in children with 22q11.2Dup, compared to children with 22q11.2DS, children with ASD and typically developing children. Consistent with the results of other studies [25,26,27], mean total SRS scores met clinical cut-offs for mild–moderate social responsiveness deficits in probands with 22q11.2Dup. Based on this study and in agreement with the results from Verbesselt et al. [18], the estimated prevalence of ASD in 22q11.2Dup is 14–25%, with another third of the sample showing autistic features.

Up to now, no studies have focused on the communication profile in children with 22q11.2Dup, and only some case reports describe the presence of speech-language problems [18,19,20,21,22]. Therefore, the purpose of this study is to characterize social-communicative behaviors in school-aged children with 22q11.2Dup. Standardized screening instruments completed by parents are seen as a suitable starting point for collecting data on social-communicative behaviors because parents are reliable informants regarding the abilities of their children [28,29,30,31]. In addition, standardized questionnaires, such as SRS-2 and Children’s Communication Checklist (CCC-2), have normed references and therefore enable comparisons with typically developing peers in the general population [32,33].

To further contribute to the characterization of syndrome-specific features in children with 22q11.2Dup, two relevant control groups were included. The first control group consists of full non-carrier siblings of children with 22q11.2Dup, providing insight into genetic and environmental background factors that may modulate language, cognitive and behavioral outcomes in children with 22q11.2Dup. Including siblings as a control group reduces the impact of contextual factors such as socio-economic status and educational attainment of the parents. The second control group consists of age-matched children with 22q11.2DS, enabling pairwise cross-CNV comparisons in a cross-sectional research design. Comparing children with reciprocal CNVs allows us to investigate whether CNVs within the same chromosomal locus have a similar impact on the phenotype or whether changes in gene dosage are associated with mirror phenotypes, as was reported for the 16p11.2 locus [34]. Additionally, cross-CNV comparisons may contribute to the identification of syndrome-specific social-communicative features [35].

The current study has a three-fold objective. First, the skills of children with 22q11.2Dup, their siblings, and children with 22q11.2DS will be compared to the norm group scores in the general population. We expect the scores of children with 22q11.2 CNVs to differ from the norm group scores, whereas scores of siblings are expected to be within the same range as the norm group scores. Second, the skills of children with the 22q11.2Dup will be compared to those of their non-carrier siblings and of age-matched children with 22q11.2DS. We hypothesize that children with 22q11.2Dup will take an intermediate position between their siblings and children with 22q11.2DS, meaning that they will probably display better social-communicative skills than children with 22q11.2DS and worse than their siblings. Finally, parental reports of de novo and familial duplications will be compared to elucidate the influence of the inheritance pattern on social-communicative skills. Likewise, sex differences and differences depending on the country of residence or comorbid ASD diagnosis will be explored.

## 2. Materials and Methods

### 2.1. Participants

This prospective study includes 49 participants, consisting of 19 unrelated children with 22q11.2Dup, 19 unrelated children with 22q11.2DS, and 11 unrelated non-carrier siblings of the children with 22q11.2Dup. All participants were school-aged children between 6 and 16 years. We only included monolingual Dutch-speaking children to avoid the impact of multilingualism on language development [36,37,38]. Prematurity was an exclusion criterium (i.e., gestational age < 37 weeks) due to the known influence on language development [39,40]. Additional exclusion criteria were no language output on the sentence level and severe sensorimotor problems such as severe hearing loss (≥55 dB *HL*) or severe visual impairments, except for cerebral visual impairment (CVI). Children with comorbid NDDs such as CVI, ASD, and ADHD were not excluded from the sample because of the high comorbidity, and in the case of ASD, traits may differ between children with ASD with and without underlying genetic defects [41]. Finally, participants with more than one (likely) pathogenic CNV were excluded.

Using a genetic-first approach, all children with 22q11.2Dup and 22q11.2DS had a confirmed diagnosis based on the fluorescent in situ hybridization technique (FISH) or micro-array (array-CGH). The majority of children with 22q11.2Dup carry the most common 3 Mb microduplication, located at LCRs A-D. One child had breakpoints situated at LCRs A-B, one at LCRs A-E, one at LCRs A-H, one at LCRs B-C, and one at LCRs C-D. All children with 22q11.2DS carry the A-D microdeletion. All children with 22q11.2 CNVs were index patients diagnosed in a clinical setting because of developmental or medical issues or a combination of both. Due to ethical considerations, siblings did not undergo genetic testing unless there was an indication to do so. However, even in familial cases, there was no indication for referral for genetic testing in siblings.

Table 1 shows demographic and clinical data for the three groups of children. Data on developmental milestones and education were obtained from digital medical records or anamnesis provided by parents. Speech-language milestones were delayed in 79% of children with 22q11.2Dup and 95% of children with 22q11.2DS. Speech-language therapy has been received by 84% of children with 22q11.2Dup, all children with 22q11.2DS, and 27% (3/11) of siblings. While all siblings follow regular education, 63% of children with 22q11.2Dup and 74% of those with 22q11.2DS attend special education.

### 2.2. Research Design

All participants were recruited through the Centre for Human Genetics of UZ Leuven or Maastricht University Medical Centre. Questionnaires were provided and completed through the online platform Qualtrics. Data were prospectively collected during home visits or consultations at the hospital from 2012 to 2022. A subgroup of children with 22q11.2DS and 8 children with 22q11.2Dup has previously been published [17,18,42].

A cross-sectional study design with pairwise comparisons was applied. The first pairwise comparison consisted of CNV pairs, for which 19 children with 22q11.2DS were matched to children with 22q11.2Dup on chronological age (CA), reducing the impact of age-related advantages such as having more experience in social interactions. Age matching was within 0.5 years of age, with an average deviation of 3 months. No significant differences between groups were found for the matching parameter using paired samples Student’s *t*-test (*t* = 0.129, *p* = 0.899) [35]. The second pairwise comparison was aimed at intrafamilial pairs, consisting of children with 22q11.2Duplication and their non-carrier siblings. Only 11 children with 22q11.2Dup had a sibling willing to participate who met the criteria of age, at term birth, and no neurological defects. In families with more than one sibling, the sibling closest in age to the child with the 22q11.2Dup was selected.

### 2.3. Measurements

Children’s social-communicative skills were investigated by means of two standardized parental questionnaires. The first one is the Dutch edition of the Children’s Communication Checklist—Second version (CCC-2) [43,44], a 70-item screening instrument, used to assess a wide scope of everyday communicative skills, including speech, structural and pragmatic language skills, and social abilities. Parents need to indicate how often their child shows certain communicative behavior on a frequency scale of 0–3 (0 = less than once a week or never, 3 = several times a day or always). Raw scores can be converted into scaled scores (SS) based on the chronological age (CA) of the participant. The questionnaire is normed for children between 4 and 15.6 years of age. Since one sibling was already 16 years of age, the scores of the oldest norm group were used to convert raw scores to scaled scores.

In total, there are 10 different norm-referenced subscales, each with an average SS of 10 and a standard deviation (SD) of 3. The higher the score, the weaker the social-communicative skills; e.g., an SS of 17 on a given subscale is more than two SDs above average, implying considerable difficulties within this domain. The first four subscales measure speech and structural language components (A. Speech, B. Syntax, C. Semantics, and D. Coherence), while the next four assess pragmatic language (E. Inappropriate initiation, F. Stereotyped language, G. Use of context, H. Non-verbal communication) and the final two focus on autistic features (I. Social relations and J. Interests). In addition, we will focus on two main composite scores: the General Communication Composite (GCC) and the Pragmatic Composite (PC). The GCC is based on all communication scales (A-H) with a clinical cut-off of 104 points (pc 10) for moderate communication problems and 117 (pc 2) for severe communication deficits, while the PC is the combined score of the four pragmatic subscales, giving an indication of pragmatic language difficulties. The cut-offs for moderate and severe pragmatic problems are scores of 53 (pc 10) and 60 (pc 2) respectively [11,44,45].

The second questionnaire is the Dutch edition of the Social Responsiveness Scales or Social Responsiveness Scales—Second edition (SRS or SRS-2) [33,46], a 65-item valid screening questionnaire that uses a Likert-scale of 1–4 (1 = not true, 4 = almost always true) to quantify deficits in social behavior associated with ASD. It contains 5 different treatment subscales: Social Awareness, Social Cognition, Social Communication, Social Motivation, and Restricted Interests and Repetitive Behavior [24,33,46,47,48]. For participants between 4 and 18 years of age, raw scores can be converted to country- and sex-normed T-scores, each with an average of 50 and SD of 10. The higher the T-score, the more social responsiveness problems someone experiences with T-scores between 61 and 75 (pc < 16) are interpreted as mild–moderate and above 75 (pc < 1.2) as severe social responsiveness impairments, according to the test manual [33].

Due to the earlier data collection, parents of children with 22q11.2DS have completed the SRS, whereas parents of children with 22q11.2Dup and their non-carrier children have filled out the SRS-2. In the Dutch version, there are no differences between SRS and SRS-2 regarding the questions and norms, apart from the addition of two composite scores, Restricted interests and behavior (RIB) and Social communication and interaction (SCI), to better correspond with the DSM-5 criteria [49]. The RIB is the same as the restricted interests and repetitive behavior subscale, while the SCI consists of the four other treatment subscales [33,47]. Both composite scores were also derived for children with 22q11.2DS. Henceforth, we will refer to SRS-2 for all groups of children.

### 2.4. Data Analysis

Due to the violation of assumptions, non-parametric (Wilcoxon signed rank) one-sample *t*-tests are applied to investigate whether the skills of children with 22q11.2 CNVs and siblings significantly differ from the norm group scores. Given the expected large intra-group variability in children with 22q11.2 CNVs, traditional statistical testing is combined with descriptive and qualitative analyses using a three-tiered method. Hence, the scores are analyzed at three different levels with statistical analyses of group differences, proportion differences across groups, and detailed characterization of typical or atypical individual patterns [50].

At the group level, CNV pairs (19 age-matched children with 22q11.2Dup and 22q11.2DS) and intrafamilial pairs (11 children with 22q11.2Dup and their siblings) are statistically compared on the main composite scores of SRS-2 and CCC-2, using pairwise Student’s *t*-tests. At the intermediate or subgroup level, the proportions of participants with clinical scores on SRS-2 and CCC-2 composite scores are compared across CNV and intrafamilial pairs using McNemar’s test. Clinical cut-off scores were 104 for GCC, 53 for PC, and 60 for SRS scores. On the subtest level, children were considered to have social-communicative difficulties when their scores deviated more than one SD from the norm group average. Consequently, subgroup analyses allow investigating whether individual variations affect the mean value of the group, which is useful in small sample studies with high risks for skewed group results by large intra-group variability. At the individual level, we look at interesting profiles within the group of children with 22q11.2Dup, such as the impact of the inheritance pattern, sex, or country of residence on the social-communicative results, using independent-sample *t*-tests. Finally, we qualitatively investigate the influence of an ASD diagnosis on outcomes. Due to multiple testing, Bonferroni corrections were applied to reduce type I errors. All statistical analyses were performed using JASP version 0.16.3 [51] and R 4.2.1 [52,53].

## 3. Results

### 3.1. CNVs and Siblings Compared to Norm Group Scores

Figure 1 depicts boxplots of the composite scores on CCC-2 and SRS-2 for the three groups of children, with the gray zones indicating how many children have mild–moderate to severe social-communicative problems and the dashed line showing norm group averages. The box plots show a wide range of scores for children with CNVs, especially in the 22q11.2Dup group. One-sample Wilcoxon signed rank *t*-tests were used to compare the three groups of children to the norm group on all reported composite scores. Results show that parental reports of children with 22q11.2Dup significantly differed from the norm group on all CCC-2 and SRS-2 composite scores (0.001 < *p* < 0.004). The same results were found in children with 22q11.2DS (*p* < 0.001). In both groups, the results remained significant after the Bonferroni correction. In contrast to both CNV groups, no significant differences were found between parental reports of siblings and the norm group on the CCC-2 and SRS-2 composite scores.

### 3.2. Cross-CNV and Intrafamilial Comparisons at Group Level: Mean Differences

Mean composite scores in Table 2 illustrate mild–moderate to severe reported social-communicative difficulties across all composite scores in children with 22q11.2DS, without reported social-communicative difficulties in children with 22q11.2Dup and their siblings, except for mean SCI scores in the 22q11.2Dup group. Parametric paired *t*-tests were used to perform cross-CNV and intrafamilial comparisons. Pairwise cross-CNV comparisons revealed significantly weaker GCC, PC, and RIB scores in children with 22q11.2DS compared to children with 22q11.2Dup. Additionally, pairwise intrafamilial comparisons showed significantly weaker scores across all CCC-2 and SRS-2 composite scores in children with 22q11.2Dup compared to their siblings. However, all former significant results did not survive Bonferroni correction.

At the subtest level, box plots in Figure 2 show similar distributions across all subtests, indicating that children with 22q11.2Dup mostly show weaker scores than their siblings and better scores than children with 22q11.2DS. For children with 22q11.2DS, mean subtest scores are within the clinical range across all subtests, while children with 22q11.2Dup only show clinical scores for SRS-2 subtests Cognition, Communication, Motivation, and Restrictive Interests/Repetitive Behaviors. Mean subtest scores of siblings are within the normal range across all subtests.

### 3.3. Cross-CNV and Intrafamilial Comparisons at Subgroup Level: Proportion Differences

Based on the CCC-2, general communication (GCC) difficulties have been reported in 47% (9/19) of children with 22q11.2Dup, 9% (1/11) of their siblings, and 79% (15/19) of children with 22q11.2DS (see Table 3). Although the reported proportions of GCC difficulties differ across groups, both intrafamilial and cross-CNV pairwise comparisons were not found to be significant according to McNemar’s test. Similarly, parents reported pragmatic problems (PC) in 37% (7/19) of children with 22q11.2Dup, 9% (1/11) of their siblings, and 68% (13/19) of children with 22q11.2DS without any significant differences for intrafamilial and cross-CNV comparisons. At the subtest level, the most commonly reported problems in children with 22q11.2Dup were problems with Speech in 58% (11/19) and Use of Context and Coherence in 53% (10/19), while in their siblings, the most commonly reported difficulties were difficulties with Speech in 27% (2/11), and Syntax and Coherence in 18% (2/11%). As in children with 22q11.2Dup, the most common concerns in children with 22q11.2DS were problems with Speech and Use of Context in 79% (15/19) and Coherence in 68% (13/19).

Based on the SRS-2, total social responsiveness problems have been reported in 47% (9/19) of children with 22q11.2Dup, 9% (1/11) of their siblings, and 79% (15/19) of children with 22q11.2DS. Proportions of difficulties on the composite scores are displayed in Table 3. Intrafamilial pairwise comparisons did not reveal any statistical differences between children with 22q11.2Dup and their siblings for SRS-2 composite scores. Neither did cross-CNV pairwise comparisons for SRS-2 SCI (*p* = 0.131) or total composite score (*p* = 0.077). However, significantly more children with 22q11.2DS (95%) were reported to have RIB difficulties compared to children with 22q11.2Dup (47%, *p* = 0.016), but the results did not remain significant after Bonferroni correction. At the subtest level, the most commonly reported difficulties in children with 22q11.2Dup were problems with Social Motivation in 53% (10/19) and Social Communication and Restricted Interests/Repetitive Behaviors in 47% (9/19), while Social Motivation is the most commonly reported problem in their siblings in 18% (2/11). In children with 22q11.2DS, the most commonly reported difficulties were Restricted Interests/Repetitive Behaviors in 95% (18/19), problems with Social Cognition in 89% (17/19), and Social Communication in 68% (13/19).

### 3.4. Within-Group Comparisons at Individual Level

Because of the specific interest in social-communication profiles of children with 22q11.2Dup, we analyzed certain subgroups in more detail. First, parental reports of children with de novo and familial duplications were compared to investigate the influence of the inheritance pattern on the reported social-communicative profile. Qualitatively, parents reported more heterogeneous profiles in children with inherited duplications compared to children with de novo duplications on SRS composite scores, but statistical tests failed to find any significant differences. Accordingly, no sex or country differences were found on the composite scores. Qualitatively, girls showed weaker SRS composite scores with higher variability compared to boys, whereas Dutch children from the Netherlands had more heterogeneous profiles with better mean GCC scores compared to Belgian Dutch-speaking children. The two children with 22q11.2Dup and comorbid ASD showed qualitatively weaker SRS composite scores, but CCC composite scores were in line with the scores in the overall 22q11.2Dup group.

## 4. Discussion

The purpose of the current study was to characterize social-communicative skills in children with 22q11.2Dup, compared to the profiles of their non-carrier siblings and age-matched children with 22q11.2DS. Moreover, we aimed to investigate whether the profiles of these groups differed from norm group profiles. Therefore, two standardized screening instruments, the CCC-2 and SRS-2, were completed by parents. Additionally, the three-tiered method was used to analyze between- and within-group differences in the composite scores of both questionnaires. Based on parental reports, children with 22q11.2 CNVs experience more social-communicative difficulties compared to their typically developing peers in the general population, which is in agreement with the literature [23,27]. In contrast, siblings of children with 22q11.2Dup in the current study did not differ from the norm group and may therefore be compared to peers in the general population.

In children with 22q11.2 CNVs, all mean SRS-2 composite scores met clinical cut-offs for mild–moderate to severe social responsiveness concerns, confirming previous research [23,25]. Comparisons at the group level demonstrated that children with 22q11.2Dup performed weaker on all aspects of social responsiveness compared to their siblings but only better on Restricted Interests/Repetitive Behaviors compared to children with 22q11.2DS. Consequently, the absence of significant differences between 22q11.2 CNVs for SCI and total SRS-2 scores may suggest that, on average, children with 22q11.2Dup have similar levels of social communication and interaction (SCI) problems as children with 22q11.2DS. Mean CCC-2 composite scores, measuring communication challenges, were just below clinical cut-offs in children with 22q11.2Dup, whereas they met clinical cut-offs for mild–moderate communication concerns in children with 22q11.2DS. Pairwise comparisons confirmed that children with 22q11.2Dup show better general communication (GCC) and pragmatic skills (PC) compared to children with 22q11.2DS but weaker overall social-communication skills compared to their siblings. These results might indicate that children with 22q11.2Dup take an intermediate position between their siblings and children with 22q11.2DS regarding communication skills. Moreover, the heterogeneous communication profiles in siblings might suggest that—in addition to the duplication—other factors, such as the broader genetic background and socio-economic status, play a role in the social-communicative outcomes [54,55].

Proportion differences at the subgroup level showed that approximately half (47%) of the children with 22q11.2Dup, 9% of their siblings, and most (79%) of the children with 22q11.2DS have general communication (GCC) and social responsiveness difficulties, with slightly lower rates for pragmatic difficulties (PC). Based on the SRS-2, no significant differences were found regarding social communication and interaction (SCI) between both CNV groups, which is in line with the results of Lin et al. [27]. Remarkably, almost all children with 22q11.2DS (95%) show Restricted Interests/Repetitive Behaviors (RIB), which is significantly higher in comparison to children with 22q11.2Dup (47%). However, results must be interpreted with caution because none of these differences remained statistically significant after the Bonferroni correction. Additionally, some of these results contrast with earlier findings [23], stating that children with 22q11.2Dup who demonstrated ASD traits without meeting all criteria for the diagnosis of ASD (*n* = 15, 4–18 years) exhibit more restricted and repetitive behaviors. Conversely, a study of 100 patients with 22q11.2DS (1–35 years) indicated that children with 22q11.2DS rather show social communication deficits than restricted and repetitive behaviors [56]. However, Lin et al. [27] found no differences in restricted and repetitive behaviors between 38 patients with 22q11.2Dup (6–61 years) and 106 with 22q11.2DS (5–49 years). These contrasting findings might be partially explained by high rates of ASD in our 22q11.2DS cohort (*n* = 8) compared to lower rates in the 22q11.2Dup cohort (*n* = 2); however, only one patient out of 19 with 22q11.2DS did not show restricted interests and repetitive behaviors. Other potential causes are the use of different measurements, different age ranges, and limited sample sizes across these studies.

Another interesting finding in the current study is that parents of children with 22q11.2 CVNs are most concerned about the same communication domains, in particular speech, use of context, and coherence, but consistently to a lesser extent in children with 22q11.2Dup. These similar concerns might suggest overlapping communicative phenotypes in 22q11.2 CNVs. However, it should be mentioned that the nature of these indicated speech problems might be different, with patients with 22q11.2DS showing more structural defects, such as cleft palate, which potentially affects speech outcomes. In contrast, speech problems in 22q11.2Dup might be characterized by more disorder-specific features and influenced by the broader familial context since this was the most reported problem among siblings. Surprisingly, 84% of children with 22q11.2Dup received speech-language therapy, although parents reported significant communication problems in only 47% of them. Potential explanations for these seemingly contradictory findings are different indications for speech-language therapy, such as SLD in certain children or overestimation of their communication skills by parents. Therefore, direct speech and language assessments in this population may clarify whether rates of reported communication problems are underestimated in children with 22q11.2Dup. Finally, detailed analyses of subgroups in children with 22q11.2Dup showed no fundamental sex differences or differences dependent on the inheritance pattern, country of residence, or comorbid ASD diagnosis.

### Strengths, Limitations, and Future

The inclusion of two relevant control groups is a key strength of the current study, suggesting an impact of the duplication in addition to the familial context on the social-communicative phenotype. Moreover, using a standardized instrument guaranteed reliable and valid norm-referenced results. Although parents are seen as reliable informants regarding everyday social-communicative skills, using an indirect approach might introduce bias, such as social desirability, misjudgments, misinterpretation, or even insufficient understanding of the questions. Therefore, questionnaires should be complemented by direct measurements in future studies to confirm the current findings in children with 22q11.2Dup and to further delineate the speech, language, and social communication profiles in this population [29,30]. Since the results did not control for cognitive abilities in cross-CNV comparisons or for age in intrafamilial comparisons, differences detected in the social-communication profile might be partly attributed to cognitive differences or more experience in everyday social-communicative interactions. Consequently, in addition to direct assessment of language, assessments of cognitive functioning are needed to determine the exact role of this potentially confounding factor.

The use of a genetic-first approach in a clinically ascertained cohort might introduce bias in the observed social-communication profiles and, therefore, not cover the whole spectrum of profiles. More likely, rates of reported problems will be lower in the population of children with 22q11.2Dup, as children without an indication for diagnosis are often not referred for genetic testing. Therefore, future studies should include a third comparison group consisting of carrier siblings diagnosed through segregation analysis. Since children with 22q11.2 CNVs did not undergo whole genome sequencing, the potential presence of additional pathogenic CNVs or single nucleotide variants might explain differences in phenotypes as well, especially in children with a rather severe phenotype. In addition, not all siblings were genetically tested; therefore, we could not exclude the presence of a CNV with certainty. However, in the remaining families, there was no indication for genetic testing of siblings; they all attended regular education, and there were no concerns regarding their development.

Two final limitations are the small sample in the current study and the heterogeneity of the population, leading to lower statistical power and restricting our ability to draw general conclusions about the whole population of children with 22q11.2Dup. Interestingly, the relatively small sample did not prevent us from finding significant results. Nevertheless, large-scale multicenter studies are needed to further delineate the social-communicative profile in this heterogeneous population. Despite its limitations, this study certainly adds to the characterization of the social-communicative skills in children with 22q11.2Dup.

## 5. Conclusions

The current study contributes to the understanding of the social-communicative phenotype in children with 22q11.2Dup, in comparison to the profiles of their siblings and age-matched children with 22q11.2DS. These results are important for healthcare professionals across different clinical settings and indicate the need for social-communicative follow-up in children with 22q11.2Dup. Since parents report high rates of social-communicative challenges in children with 22q11.2Dup, healthcare professionals should be aware of the high risk of social-communicative problems and refer to a speech-language pathologist for screening or diagnostic testing. Finally, future research should focus on deep phenotyping of the communication profile of children with 22q11.2Dup using standardized direct language assessments.

## Figures and Tables

**Figure 1 genes-13-01801-f001:**
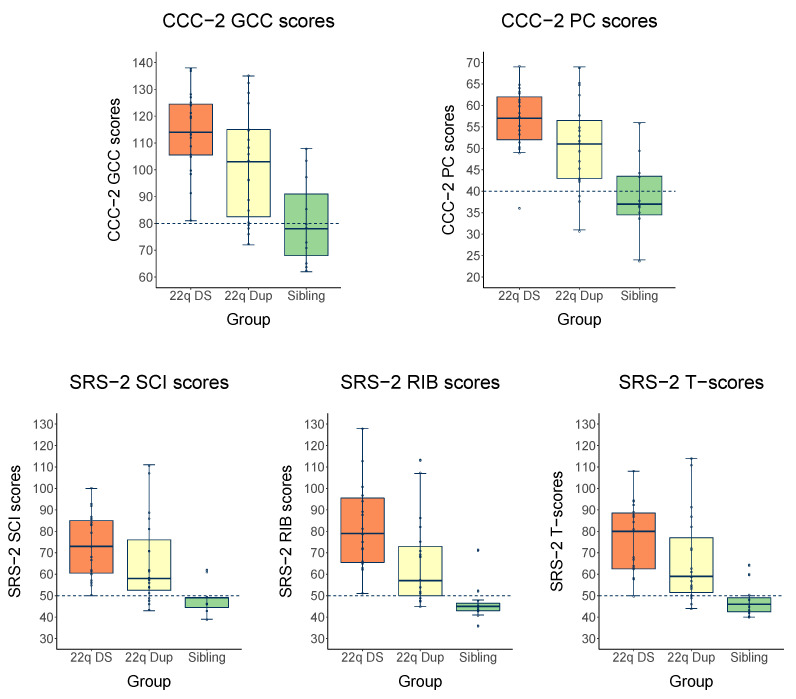
Boxplots for CCC-2 and SRS-2 composite scores across groups. The dashed lines show norm group averages. The gray zones indicate the severity of the problems; the darker the gray, the more severe the difficulties: mild–moderate = light gray zone and severe = darker gray zone, based on clinical cut-off scores for CCC-2 and SRS-2. Abbreviations. GCC, General Communication Composite (norm group average = 80, cut-off: >104 = mild–moderate (pc 10), ≥117 = severe (pc 2)); PC, Pragmatic Composite (norm group average = 50, cut-off: >53 = mild–moderate (pc 10), ≥60 = severe (pc 2)); SCI, Social Communication and Interaction, RIB, Repetitive interests and behavior, Total (norm group average = 50, cut-off: >60 = mild–moderate (pc 16), ≥76 severe (pc 1.8)).

**Figure 2 genes-13-01801-f002:**
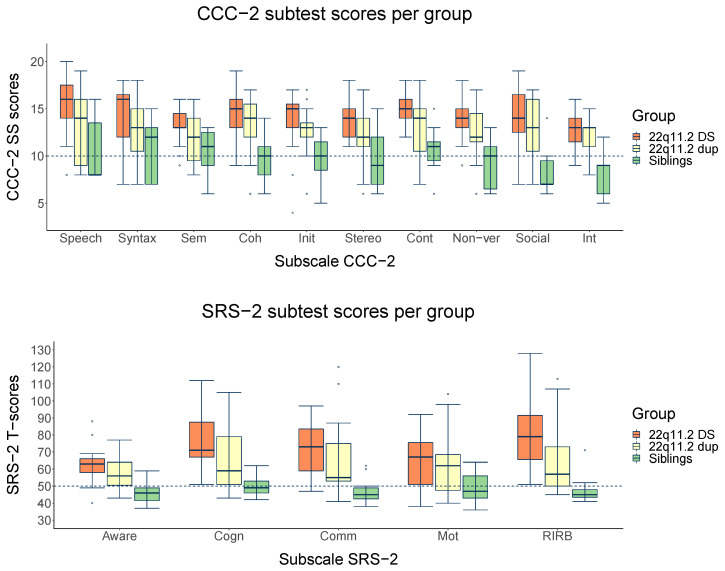
Boxplots for CCC-2 (M = 10, SD = 3) and SRS-2 (M = 50, SD = 10) group scores across subtests. The dashed lines show norm group averages. The gray zones indicate the severity of the problems; the darker the gray, the more severe the difficulties: mild–moderate = light gray zone and severe = darker gray zone. Abbreviations: CCC-2 subtests: Speech; Syntax; Sem, Semantics; Coh, Coherence; Init, Inappropriate Initiation; Stereo, Stereotyped Language; Cont, Use of Context; Non-ver, Non-verbal Communication; Social, Social relations; Int, Interests. SRS-2 subtests: Aware, Social Awareness; Cogn, Social Cognition; Comm, Social Communication; Mot, Social Motivation; Aut, Restrictive Interests and Repetitive Behaviors.

**Table 1 genes-13-01801-t001:** Demographic and clinical characteristics across groups.

	22q11.2DS	22q11.2Dup	Siblings of Dup
Sample Size (*n*)	19	19	11
Sex (*n*, %)			
Male	14 (74%)	10 (53%)	4 (36%)
Female	5 (26%)	9 (47%)	7 (64%)
Chronological age (years·mo)			
Average (SD)	10.7 (2.5)	10.7 (2.5)	10.10 (2.10)
Median	11.2	11	11
Range	6.7–14.4	6.8–14.9	6.3–16.1
Country of residence (*n*, %)			
Belgium	15 (79%)	10 (53%)	5 (45%)
The Netherlands	4 (21%)	9 (47%)	6 (55%)
Type of education (*n*, %)			
Special education	14 (74%)	12 (63%)	0 (0%)
Regular education	5 (26%)	7 (37%)	11 (100%)
Speech-language delays (*n*, %)	18 (95%)	15 (79%)	0 (0%)
Speech-language therapy (*n*, %)	19 (100%)	16 (84%)	3 (27%)
Formal NDD diagnoses (*n*, %)			
ASD	8 (42%)	2 (11%)	0 (0%)
ADHD	4 (21%)	4 (21%)	0 (0%)
SLD	1 (5%)	4 (21%)	0 (0%)
DCD	0 (0%)	4 (21%)	0 (0%)
DLD	0 (0%)	3 (16%)	0 (0%)
CVI	0 (0%)	3 (16%)	0 (0%)
Inheritance pattern (*n*, %)			
De novo	18 (95%)	8 (42%)	
Inherited	1 (5%)	8 (42%)	/
Unknown	0 (0%)	3 (16%)	

Abbreviations: NDD, neurodevelopmental disorders; ASD, autism spectrum disorder; ADHD, attention deficit/hyperactivity disorder; SLD, specific learning disorder; DLD, developmental language disorder; CVI, cerebral visual impairment; DCD, developmental coordination disorder.

**Table 2 genes-13-01801-t002:** Mean composite results on CCC-2 and SRS-2 for cross-CNV and intrafamilial comparisons.

	22q11.2Dup (*n* = 19)	22q11.2DS (*n* = 19)	*t*-Test *t* =, *p* =	22q11.2Dup (*n* = 11)	Siblings Dup (*n* = 11)	*t*-Test *t* =, *p* =
CCC-2 GCC Mean (SD)	101.58 (20.29)	114.12 (14.99)	*t* = −2.281	98.36 (19.67)	80.55 (16.01)	*t* = 2.647
Range	72.00–135.00	81.00–138.00	*p* = 0.035 *	72.00–132.00	62.00–108.00	*p* = 0.024 *
CCC-2 PC Mean (SD)	50.58 (10.25)	56.63 (7.59)	*t* = −2.190	49.27 (8.79)	39.09 (8.57)	*t* = 3.136
Range	31.00–69.00	36.00–69.00	*p* = 0.042 *	38.00–65.00	24.00–56.00	*p* = 0.011 *
SRS-2 SCI Mean (SD)	66.05 (20.07)	73.47 (15.09)	*t* = −1.313	58.18 (13.00)	48.36 (7.50)	*t* = 2.276
Range	43.00–111.00	50.00–100.00	*p* = 0.206	43.00–89.00	39.00–62.00	*p* = 0.046 *
SRS-2 RIB Mean (SD)	65.53 (19.94)	82.05 (19.97)	*t* = −2.327	57.27 (12.33)	46.64 (9.00)	*t* = 2.241
Range	45.00–113.00	51.00–128.00	*p* = 0.032 *	45.00–86.00	36.00–71.00	*p* = 0.049 *
SRS-2 Total Mean (SD)	66.79 (21.06)	76.05 (16.28)	*t* = −1.508	58.46 (13.46)	47.82 (7.79)	*t* = 2.292
Range	44.00–114.00	50.00–108.00	*p* = 0.149	44.00–91.00	40.00–64.00	*p* = 0.045 *

Alpha = 0.05 *; alpha after Bonferroni correction = 0.01. GCC, General Communication Composite (norm group average = 80, cut-off: >104 = mild–moderate, ≥117 = severe); PC, Pragmatic Composite (norm group average = 40, cut-off: >53 = mild–moderate, ≥60 = severe); SCI, Social Communication and Interaction, RIB, Restricted Interests and Repetitive Behavior, Total (norm group average = 50, cut-off: >60 = mild–moderate, ≥76 severe).

**Table 3 genes-13-01801-t003:** Proportions of children with reported difficulties across composite scores on CCC-2 and SRS-2.

	22q11.2DS (*n* = 19)	22q11.2Dup (*n* = 19)	Siblings of Dup (*n* = 11)
CCC-2 GCC	15/19 (79%)	9/19 (47%)	1/11 (9%)
CCC-2 PC	13/19 (68%)	7/19 (37%)	1/11 (9%)
SRS-2 SCI	14/19 (74%)	9/19 (47%)	2/11 (18%)
SRS-2 RIB	18/19 (95%)	9/19 (47%)	1/11 (9%)
SRS-2 Total	15/19 (79%)	9/19 (47%)	1/11 (9%)

GCC, General Communication Composite (cut-off: >104 = mild–moderate problems); PC, Pragmatic Composite (cut-off: >53 = mild–moderate problems); SCI, Social Communication and Interaction, RIB, Restricted Interests and Repetitive Behavior, Total (cut-off: >60 = mild–moderate problems).

## Data Availability

The datasets used and/or analyzed during the current study are available from the corresponding author upon reasonable request.

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
