# Peer review of "Parent-Reported Social-Communicative Skills of Children with 22q11.2 Copy Number Variants and Siblings"

_genes, 2022, doi:10.3390/genes13101801_

Round 1
Reviewer 1 Report
The article presents the results of a study aimed at investigating relational and communicative skills among children with 22q11.2 duplication in order to better characterize their phenotype. This purpose is valuable given the literature reporting an increased risk of neurodevelopmental disorders in copy number variations.
Overall, the paper is interesting, well-written and well-referenced. The methodology is robust, and the instruments employed to assess communication difficulties are well validated and both presents relevant normative values for comparison. The choice to include two matched control groups (age-matched children with 22q11.2 deletion and age-matched non-carrier siblings is remarkable and make it possible to control for relevant influences such as the family environment and the shared context.
The results are clearly presented, and their implications are thoroughly discussed with appropriate references to the literature. Furthermore, the authors did a great job in highlighting both the strengths and the limitations of their study. The accuracy of the research design compensates for the small sample size, which is indeed understandable given the inclusion criteria and the occurrence of 22q11.2 duplication.
A couple of minor sidenotes: the response format described in line 166 is not properly a Likert scale, and could be better described as a frequency scale. In line 203, "DSM-5" should be used instead of "DSM-V". Finally, the choice to use Bonferroni correction for multiple comparisons could be excessively conservative given the exploratory nature of the study and its limited statistical power. An alternative such as Benjamini–Hochberg procedure (false discovery rate) could have been considered. However, since the authors correctly reported both the original p-values and the corrected p-values for each statistical test, I definitely consider their decision to be fair.
Author Response
Dear reviewer,
Thank you for giving us the opportunity to revise and resubmit our manuscript. We appreciate your time and effort in providing us with insightful comments.
We address each of the questions or remarks of the reviewer in the order they were provided in the commentary document.
We believe that the quality of the manuscript has substantially been improved and we look forward to receiving your feedback on the revised manuscript and responses at your earliest convenience.
Sincerely,
The authors.
Point 1: The response format described in line 166 is not properly a Likert scale, and could be better described as a frequency scale.
Response 1: We agree with the comment of the reviewer and revised accordingly.
Point 2: In line 203, "DSM-5" should be used instead of "DSM-V".
Response 2: We agree with the comment of the reviewer and revised accordingly.
Point 3: Finally, the choice to use Bonferroni correction for multiple comparisons could be excessively conservative given the exploratory nature of the study and its limited statistical power. An alternative such as Benjamini–Hochberg procedure (false discovery rate) could have been considered. However, since the authors correctly reported both the original p-values and the corrected p-values for each statistical test, I definitely consider their decision to be fair.
Response 3: Thank you for this valuable feedback. We recognize the conservative nature of Bonferroni correction and decided therefore to report both the original p-values and the corrected p-values for each statistical test. Since it is considered a fair approach, we have not replaced Bonferroni correction with Benjamini-Hochberg procedure.

Reviewer 2 Report
As the author mention, the strength of the study is the inclusion of the siblings of the affected siblings. The limitation is the small cohort.
I have just a few minor comments:
Abstract line 25 "of 22q11 dups"-there is a typo
Line 310-325 it would be clearer to present the numbers in the table
Line 390-how big were other cohorts? Please give an example numbers
Author Response
Dear reviewer,
Thank you for giving us the opportunity to revise and resubmit our manuscript. We appreciate your time and effort in providing us with insightful comments.
We address each of the questions or remarks of the reviewer in the order they were provided in the commentary document.
We believe that the quality of the manuscript has substantially been improved and we look forward to receiving your feedback on the revised manuscript and responses at your earliest convenience.
Sincerely,
The authors.
Point 1: Abstract line 25 "of 22q11 dups"-there is a typo.
Response 1: The typo has been corrected.
Point 2: Line 310-325 it would be clearer to present the numbers in the table
Response 2: We agree with the comment of the reviewer and added the following table, consisting of the percentages of children with considerable problems across the five composite scores of CCC-2 and SRS-2.
Table 3. Proportions of children with reported difficulties across composite scores on CCC-2 and SRS-2.
|
|
22q11.2DS (n=19) |
22q11.2Dup (n=19) |
Siblings of Dup (n=11) |
|
CCC-2 GCC |
15/19 (79%) |
9/19 (47%) |
1/11 (9%) |
|
CCC-2 PC |
13/19 (68%) |
7/19 (37%) |
1/11 (9%) |
|
SRS-2 SCI |
14/19 (74%) |
9/19 (47%) |
2/11 (18%) |
|
SRS-2 RIB |
18/19 (95%) |
9/19 (47%) |
1/11 (9%) |
|
SRS-2 Total |
15/19 (79%) |
9/19 (47%) |
1/11 (9%) |
* GCC, General Communication Composite (cut-off: > 104 = mild-moderate problems); PC, Pragmatic Composite (cut-off: > 53 = mild-moderate problems); SCI, Social Communication and Interaction, RIB, Restricted interests and repetitive behavior, Total (cut-off: > 60 = mild-moderate problems).
Point 3: Line 390-how big were other cohorts? Please give an example numbers
Response 3: We revised accordingly and added the following numbers in the text.
Additionally, some of these results contrast with earlier findings [24], stating that children with 22q11.2Dup who demonstrated ASD traits without meeting all criteria for the diagnosis of ASD (n=15, 4-18 years) exhibit more restricted and repetitive behaviors. Conversely, a study of 100 patients with 22q11.2DS (1-35 years) indicated that children with 22q11.2DS rather show social communication deficits than restricted and repetitive behaviors [57]. However, Lin et al. [26] found no differences regarding restricted and repetitive behaviors between 38 patients with 22q11.2Dup (6-61 years) and 106 with 22q11.2DS (5-49 years). These contrasting findings might be partially explained by high rates of ASD in our 22q11.2DS cohort (n=8) compared to lower rates in the 22q11.2Dup cohort (n=2), however, only one patient out of 19 with 22q11.2DS did not show restricted interests and repetitive behaviors. Other potential causes are the use of different measurements, different age ranges and limited sample sizes across these studies.
